# Disentangled Generative Graph Representation Learning

## Abstract

Recently, generative graph models have shown promising results in learning graph representations through self-supervised methods. However, most existing generative graph representation learning (GRL) approaches rely on random masking across the entire graph, which overlooks the entanglement of learned representations. This oversight results in non-robustness and a lack of explainability. Furthermore, disentangling the learned representations remains a significant challenge and has not been sufficiently explored in GRL research. Based on these insights, this paper introduces **DiGGR** (**Di**sentangled **G**enerative **G**raph **R**epresentation Learning), a self-supervised learning framework. DiGGR aims to learn latent disentangled factors and utilizes them to guide graph mask modeling, thereby enhancing the disentanglement of learned representations and enabling end-to-end joint learning. Extensive experiments on 11 public datasets for two different graph learning tasks demonstrate that DiGGR consistently outperforms many previous self-supervised methods, verifying the effectiveness of the proposed approach.

## 1 Introduction

Self-supervised learning (SSL) has received much attention due to its appealing capacity for learning data representation without label supervision. While contrastive SSL approaches are becoming increasingly utilized on images [Chen et al., 2020] and graphs [You et al., 2020], generative SSL has been gaining significance, driven by groundbreaking practices such as BERT for language [Devlin et al., 2018], BEiT [Bao et al., 2021], and MAE [He et al., 2022a] for images. Along this line, there is a growing interest in constructing generative SSL models for other modalities, such as graph masked autoencoders (GMAE). Generally, the fundamental concept of GMAE [Tan et al., 2022] is to utilize an autoencoder architecture to reconstruct input node features, structures, or both, which are randomly masked before the encoding step. Recently, various well-designed GMAEs have emerged , achieving remarkable results in both node classification and graph classification [Hou et al., 2022, Tu et al., 2023, Tian et al., 2023].

Despite their significant achievements, most GMAE approaches typically treat the entire graph as holistic, ignoring the graph's latent structure. As a result, the representation learned for a node tends to encapsulate the node's neighborhood as a perceptual whole, disregarding the nuanced distinctions between different parts of the neighborhood [Ma et al., 2019, Li et al., 2021, Mo et al., 2023]. For example, in a social network $G$, individual $n$ is a member of both a mathematics group and several sports interest groups. Due to the diversity of these different communities, she may exhibit different characteristics when interacting with members from various communities. Specifically, the information about the mathematics group may be related to her professional research, while the information about sports clubs may be associated with her hobbies. However, the existing approach overlooks the heterogeneous factors of node $n$, failing to identify and disentangle these pieces of

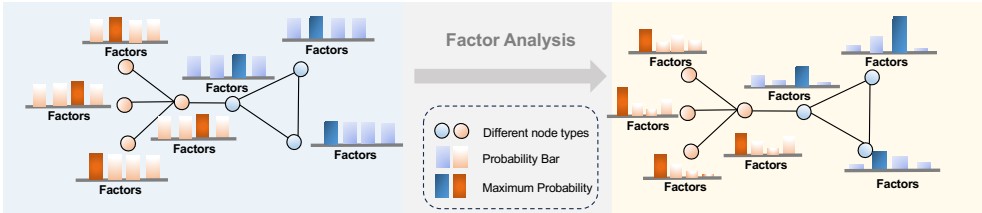

(a) Applying previous methods to GMAE          (b) Latent factor learning

Figure 1: The number of latent factors is set to 4. In Fig. 1(a), the probabilities of nodes belonging to different latent groups are similar, resulting in nodes of the same type being incorrectly assigned to different factors. In contrast, Fig. 1(b) shows that the probabilities of node-factor affiliation are more discriminative, correctly categorizing nodes of the same type into the same latent group.

information effectively [Hou et al., 2022]. Consequently, the learned features may be easily influenced by irrelevant factors, resulting in poor robustness and difficulty in interpretation.

To alleviate the challenge described above, there is an increasing interest in disentangled graph representation learning [Bengio et al., 2013, Li et al., 2021, Ma et al., 2019, Mo et al., 2023, Xiao et al., 2022], which aims at acquiring representations that can disentangle the underlying explanatory factors of variation in the graph. Specifically, many of these methods rely on a latent factor detection module, which learns the latent factors of each node by comparing node representations with various latent factor prototypes. By leveraging these acquired latent factors, these models adeptly capture factor-wise graph representations, effectively encapsulating the latent structure of the graph. Despite significant progress, few studies have endeavored to adapt these methods to to generative graph representation learning methods, such as GMAE. This primary challenge arises from the difficulty of achieving convergence in the latent factor detection module under the generative training target, thus presenting obstacles in practical implementation. As shown in Fig.1(a), directly applying the previous factor learning method to GMAE would make the factor learning module difficult to converge, resulting in undistinguished probabilities and misallocation of similar nodes to different latent factor groups.

To address these challenges, we introduce **Di**sentangled **G**enerative **G**raph **R**epresentation Learning (**DiGGR**), a self-supervised graph generation representation learning framework. Generally speaking, DiGGR learns how to generate graph structures from latent disentangle factors $z$ and leverages this to guide graph mask reconstruction, while enabling end-to-end joint learning. Specifically, $i$) To capture the heterogeneous factors in the nodes, we introduce the latent factor learning module. This module models how edges and nodes are generated from latent factors, allowing graphs to be factorized into multiple disentangled subgraphs. $ii$) To learn a deeper disentangled graph representation, we design a factor-wise self-supervised graph representation learning framework. For each subgraph, we employ a distinct masking strategy to learn an improved factor-specific graph representation. Evaluation shows that the proposed framework can achieve significant performance enhancement on various node and graph classification benchmarks.

The main contributions of this paper can be summarized as follows:

- We utilized the latent disentangled factor to guide mask modeling. A probabilistic graph generation model is employed to identify the latent factors within a graph, and it can be jointly trained with GMAE through variational inference.
- Introducing **DiGGR** (**Di**sentangled **G**enerative **G**raph **R**epresentation Learning) to further capture the disentangled information in the latent factors, enhancing the disentanglement of the learned node representations.
- Empirical results show that the proposed DiGGR outperforms many previous self-supervised methods in various node- and graph-level classification tasks.

## 2 Related works

**Graph Self-Supervised Learning:**     Graph SSL has achieved remarkable success in addressing label scarcity in real-world network data, mainly consisting of contrastive and generative methods. Con-

trastive methods, includes feature-oriented approaches[Hu et al., 2019, Zhu et al., 2020, Veličković et al., 2018], proximity-oriented techniques [Hassani and Khasahmadi, 2020, You et al., 2020], and graph-sampling-based methods [Qiu et al., 2020]. A common limitation across these approaches is their heavy reliance on the design of pretext tasks and augmentation techniques. Compared to contrastive methods, generative methods are generally simpler to implement. Recently, to tackle the challenge of overemphasizing neighborhood information at the expense of structural information [Hassani and Khasahmadi, 2020, Veličković et al., 2018], the Graph Masked Autoencoder (GMAE) has been proposed. It applies a masking strategy to graph structure [Li et al., 2023a], node attributes [Hou et al., 2022], or both [Tian et al., 2023] for representation learning. Unlike most GMAEs, which employ random mask strategies, this paper builds disentangled mask strategies.

**Disentangled Graph Learning:**   Disentangled representation learning aims to discover and isolate the fundamental explanatory factors inherent in the data [Bengio et al., 2013]. Existing efforts in disentangled representation learning have primarily focused on computer vision [Higgins et al., 2017, Jiang et al., 2020]. Recently, there has been a surge of interest in applying these techniques to graph-structured data [Li et al., 2021, Ma et al., 2019, Mercatali et al., 2022, Mo et al., 2023]. For example, DisenGCN [Ma et al., 2019] utilizes an attention-based methodology to discriminate between distinct latent factors, enhancing the representation of each node to more accurately reflect its features across multiple dimensions. DGCL [Li et al., 2021] suggests learning disentangled graph-level representations through self-supervision, ensuring that the factorized representations independently capture expressive information from various latent factors. Despite the excellent results achieved by the aforementioned methods on various tasks, these methods are difficult to converge in generative graph SSL, as we demonstrated in the experiment of Table.3. Therefore, this paper proposes a disentangled-guided framework for generative graph representation learning, capable of learning disentangled representations in an end-to-end self-supervised manner.

# 3   Proposed Method

In this section, we propose **DiGGR** (**Di**sentangled **G**enerative **G**raph **R**epresentation Learning) for self-supervised graph representation learning with mask modeling. The framework was depicted in Figure 2, comprises three primary components: Latent Factor Learning (Section 3.2), Graph Factorization (Section 3.2) and Disentangled Graph Masked autoencder (Section 3.3). Before elaborating on them, we first show some notations.

## 3.1   Preliminaries

A graph $G$ can be represented as a multi-tuple $\mathcal{G} = \{V, A, X\}$ with $N$ nodes and $M$ edges, where $|V| = N$ is the node set, $|A| = M$ is the edge set, and $X \in \mathbb{R}^{N \times L}$ is the feature matrix for $N$ nodes with $L$ dimensional feature vector. The topology structure of graph $G$ can be found in its adjacency matrix $A \in \mathbb{R}^{N \times N}$. $z \in \mathbb{R}^{N \times K}$ is the latent disentangled factor matrix, and $K$ is the predefined factor number. Since we aim to obtain the $z$ to guide the mask modeling, we first utilize a probabilistic graph generation model to factorize the graph before employing the mask mechanism. Given the graph $G$, it is factorized into $\{G_1, G_2, ..., G_K\}$, and each factor-specific graph $G_k$ consists of its factor-specific edges $A^{(k)}$, node set $V^{(k)}$ and node feature matrix $X^{(k)}$. Other notations will be elucidated as they are employed.

## 3.2   Latent Factor Learning

In this subsection, we describe the latent factor learning method. In this phase, our objective is to derive factor-specific node sets $\{V^{(1)}, V^{(2)}, ..., V^{(K)}\}$ and adjacency matrices $\{A^{(1)}, A^{(2)}, ..., A^{(K)}\}$, serving as basic unit of the graph to guide the subsequent masking. The specific approach involves modeling the distribution of nodes and edges, utilizing the generative process developed in EPM [Zhou, 2015]. The generative process of EPM under the Bernoulli-Poisson link [Zhou, 2015] can be described as:

$$\mathrm{M}_{uv} \sim \mathrm{Poisson}(\sum\nolimits_{k=1}^{K} \gamma_k z_{uk} z_{vk}), \;\; z_{uk} \sim \mathrm{Gamma}\left(\alpha, \beta\right), u, v \in [1, N] \tag{1}$$

where $K$ is the predefined number of latent factors, and $u$ and $v$ are the indexes of the nodes. Here, $\mathrm{M}_{uv}$ is the latent count variable between node $u$ and $v$; $\gamma_k$ is a positive factor activation level indicator, which measures the node interaction frequency via factor $k$; $z_{uk}$ is a positive latent variable for node

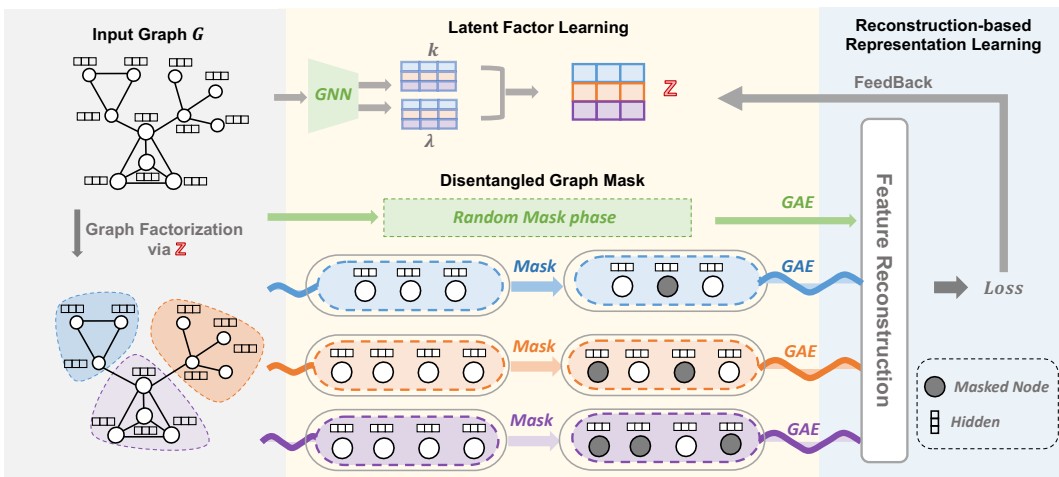

Figure 2: The overview of proposed DiGGR's computation graph. The input data successively passes three modules described in Sections 3.2 and 3.3: Latent Factor Learning, Graph Factorization, and Disentangled Graph Mask Autoencoder. Graph information will be first processed through Latent Factor Learning and Graph Factorization, the former processed the input graph to get the latent factor $z$; the latter performs graph factorization via $z$, such that in each factorized subgraph, nodes exchange more information with intensively interacted neighbors. Hence, during the disentangled graph masking phase, we will individually mask each factorized subgraph to enhance the disentanglement of the obtained node representations.

$u$, which measures how strongly node $u$ is affiliated with factor $k$. The prior distribution of latent factor variable $z_{uk}$ is set to Gamma distribution, where $\alpha$ and $\beta$ are normally set to 1. Therefore, the intuitive explanation for this generative process is that, with $z_{uk}$ and $z_{vk}$ measuring how strongly node $u$ and $v$ are affiliated with the $k$-th factor, respectively, the product $\gamma_k z_{uk} z_{vk}$ measures how strongly nodes $u$ and $v$ are connected due to their affiliations with the $k$-th factor.

**Node Factorization:** Equation 1 can be further augmented as follows:

$$\mathrm{M}_{uv} = \sum_k^K \mathrm{M}_{ukv}, \ \ \mathrm{M}_{ukv} \sim \mathrm{Poisson}\left(\gamma_k z_{uk} z_{vk}\right) \tag{2}$$

where $\mathrm{M}_{ukv}$ represents how often nodes $u$ and $v$ interact due to their affiliations with the $k$-th factor. To represent how often node $u$ is affiliated with the $k$-th factor, we further introduce the latent count $\mathrm{M}_{uk\cdot} = \sum_{v \neq u} \mathrm{M}_{ukv}$. Then, we can soft assign node $u$ to multiple factors in $\{k : \mathrm{M}_{uk\cdot}\} \geq 1$, or hard assign node $u$ to a single factor using $\arg\max_k (\mathrm{M}_{uk\cdot})$. However, our experiments show that soft assignment method results in significant overlap among node sets from different factor group, diminishing the distinctiveness. Note that previous study addressed a similar issue by selecting the top-k most attended regions [Kakogeorgiou et al., 2022]. Thus, we choose the hard assign strategy to factorize the graph node set $V$ graph into factor-specific node sets $\{V^{(1)}, V^{(2)}, \cdots, V^{(K)}\}$.

**Edge Factorization:** To create factor-specific edges $A^{(k)}$ for a factor-specific node set $V^{(k)}$, a straightforward method involves removing all external nodes connected to other factor groups. This can be defined as:

$$A_{uv}^{(k)} = \begin{cases} A_{uv}, \ \forall \, u, v \in V^{(k)}; \ u, v \in [1, N]; \\ 0, \ \ \exists \, u, v \notin V^{(k)}; \ u, v \in [1, N]. \end{cases} \tag{3}$$

Besides, the global graph edge $A$ can also be factorized into positive-weighted edges [He et al., 2022b] for each latent factor as:

$$A_{uv}^{(k)} = A_{uv} \cdot \frac{\exp\left(\gamma_k z_{uk} z_{vk}\right)}{\sum_{k'} \exp\left(\gamma_{k'} z_{uk'} z_{vk'}\right)}; \ \ k \in [1, K], u, v \in [1, N]. \tag{4}$$

Applying Equation 4 to all pairs of nodes yields weighted adjacency matrices $\{A^{(k)}\}_{k=1}^k$, with $A^{(k)}$ corresponding to latent factor $z_k$. Note that $A^{(k)}$ has the same dimension as $A$ and Equation 4

147 presents a trainable weight for each edge, which can be jointly optimized through network training,
148 showcasing an advantage over Equation 3 in this aspect. Therefore, we apply Equation 4 for edge
149 factorization.

150 **Variational Inference:**  The latent factor variable $z$ determines the quality of node and edge factor-
151 ization, so we need to approximate its posterior distribution. Denoting $z_u = (z_{u1}, ..., z_{uK}), z_u \in \mathbb{R}_+^K$,
152 which measures how strongly node $u$ is affiliated with all the $K$ latent factors, we adopt a Weibull
153 variational graph encoder [Zhang et al., 2018, He et al., 2022b]:

$$q(z_u \mid A, X) = \text{Weibull}(k_u, \lambda_u), \quad (k_u, \lambda_u) = \text{GNN}_{\text{EPM}}(A, X), \quad u \in [1, N] \tag{5}$$

154 where $\text{GNN}_{\text{EPM}}(\cdot)$ stands for graph neural networks, and we select a two-layer Graph Convolution
155 Networks (*i.e.*, GCN [Kipf and Welling, 2016a]) for our models; $k_u, \lambda_u \in \mathbb{R}_+^K$ are the shape and
156 scale parameters of the variational Weibull distribution, respectively. The latent variable $z_u$ can be
157 conveniently reparameterized as:

$$z_u = \lambda_u(-\ln(1 - \varepsilon))^{1/k_u}, \quad \varepsilon \sim \text{Uniform}(0, 1). \tag{6}$$

158 The optimization objective of latent factor learning phase can be achieved by maximizing the evidence
159 lower bound (ELBO) of the log marginal likelihood of edge $\log p(A)$, which can be computed as:

$$\mathcal{L}_z = \mathbb{E}_{q(Z \mid A,X)}[\ln p(A \mid Z)] - \sum_{u=1}^{N} \mathbb{E}_{q(z_u \mid A,X)}\left[\ln \frac{q(z_u \mid A, X)}{p(z_u)}\right] \tag{7}$$

160 where the first term is the expected log-likelihood or reconstruction error of edge, and the second
161 term is the Kullback–Leibler (KL) divergence that constrains $q(z_u)$ to be close to its prior $p(z_u)$. The
162 analytical expression for the KL divergence and the straightforward reparameterization of the Weibull
163 distribution simplify the gradient estimation of the ELBO concerning the decoder parameters and
164 other parameters in the inference network.

## 3.3  Disentangled Grpah Masked Autoencoder

166 With the latent factor learning phase discussed in 3.2, the graph can be factorized into a series of
167 factor-specific subgraphs $\{G_1, G_2, ..., G_K\}$ via the latent factor $z$. To incorporate the disentangled
168 information encapsulated in $z$ into the graph masked autoencoder, we proposed Disentangled Graph
169 Masked Autoencoder in this section. Specifically, this section will first introduce the latent factor-wise
170 GMAE and the graph-level GMAE.

### 3.3.1  Latent Factor-wise Grpah Masked Autoencoder

172 To capture disentangled patterns within the latent factor $z$ , for each latent subgraph
173 $\mathcal{G}_k = (V^{(k)}, A^{(k)}, X^{(k)})$, the latent factor-wise GMAE can be described as:

$$H_d^{(k)} = \text{GNN}_{\text{enc}}(A^{(k)}, \bar{X}^{(k)}), \tilde{X}^d = \text{GNN}_{\text{dec}}(A, H_d). \tag{8}$$

174 where $\bar{X}^{(k)}$ is the masked node feature matrix for the $k$-th latent factor, and $\tilde{X}^d$ denotes the recon-
175 structed node features. $\text{GNN}_{\text{enc}}(.)$ and $\text{GNN}_{\text{dec}}(.)$ are the graph encoder and decoder, respectively;
176 $H_d^{(k)} \in \mathbb{R}^{N \times D}$ are factor-wise hidden representations, and $H_d = H_d^{(1)} \oplus H_d^{(2)} \cdots \oplus H_d^{(K)}$. After the
177 concatenation operation $\oplus$ in feature dimension, the multi factor-wise hidden representation becomes
178 $H_d \in \mathbb{R}^{N \times (K \cdot D)}$, which is used as the input of $\text{GNN}_{\text{dec}}(.)$.

179 Regarding the mask opeartion, we uniformly random sample a subset of nodes $\bar{V}^{(k)} \in V^{(k)}$ and
180 mask each of their features with a mask token, such as a learnable vector $X_{[M]} \in \mathbb{R}^d$. Thus, the node
181 feature in the masked feature matrix can be defined as:

$$\bar{X}_i^{(k)} = \begin{cases} X_{[M]}; & v_i \in \bar{V}^{(k)} \ ; \\ X_i & ; \ v_i \notin \bar{V}^{(k)}. \end{cases} \tag{9}$$

182 The objective of latent factor-wise GMAE is to reconstruct the masked features of nodes in $\bar{V}^{(k)}$
183 given the partially observed node signals $\bar{X}^{(k)}$ and the input adjacency matrix $A^{(k)}$. Another crucial
184 component of the GMAE is the feature reconstruction criterion, often used in language as cross-
185 entropy error [Devlin et al., 2018] and in the image as mean square error [He et al., 2022a]. However,

texts and images typically involve tokenized input features, whereas graph autoencoders (GAE) do not have a universal tokenizer. We adopt the scored cosine error of GraphMAE [Hou et al., 2022] as the loss function. Generally, given the original feature $X^{(k)}$ and reconstructed node feature $\tilde{X}^{(k)}$, the defined SCE is:

$$\mathcal{L}_{\text{D}} = \frac{1}{|\bar{V}|} \sum_{i \in \bar{V}} \left(1 - \frac{X_i^T \tilde{X}_i^d}{\|X_i\| \cdot \|\tilde{X}_i^d\|}\right)^{\gamma}, \ \ \gamma \geq 1 \tag{10}$$

where $\bar{V} = \bar{V}^{(1)} \cup \bar{V}^{(2)}... \cup \bar{V}^{(K)}$ and Equation 10 are averaged over all masked nodes.The scaling factor $\gamma$ is a hyper-parameter adjustable over different datasets. This scaling technique could also be viewed as adaptive sample reweighting, and the weight of each sample is adjusted with the reconstruction error. This error is also famous in the field of supervised object detection as the focal loss [Lin et al., 2017].

**Graph-level Graph Mask Autoencoder:** For the node classification task, we have integrated graph-level GMAE into DiGGR. We provide a detailed experimental analysis and explanation for this difference in Appendix A.1.2. The graph-level masked graph autoencoder is designed with the aim of further capturing the global patterns, which can be designed as:

$$H_g = \text{GNN}_{\text{enc}}(A, \bar{X}), \ \ \tilde{X}^g = \text{GNN}_{\text{dec}}(A, H_g). \tag{11}$$

$\bar{X}$ is the masked node feature matrix, whose mask can be generated by uniformly random sampling a subset of nodes $\tilde{V} \in V$, or obtained by concatenating the masks of all factor-specific groups $\tilde{V} = \bar{V}^{(1)} \cup \bar{V}^{(2)}... \cup \bar{V}^{(K)}$. The global hidden representation encoded by $\text{GNN}_{\text{enc}}(.)$ is $H_g$, which is then passed to the decoder. Similar to Equation 10, we can define the graph-level reconstruct loss as:

$$\mathcal{L}_{\text{G}} = \frac{1}{|\tilde{V}|} \sum_{i \in \tilde{V}} (1 - \frac{X_i^T \tilde{X}_i^g}{\|X_i\| \cdot \|\tilde{X}_i^g\|})^{\gamma}, \ \ \gamma \geq 1. \tag{12}$$

which is averaged over all masked nodes.

## 3.4  Joint Training and Inference

Benefiting from the effective variational inference method, the proposed latent factor learning and dsientangled graph masked autoencoder can be jointly trained in one framework. We combine the aforementioned losses with three mixing coefficient $\lambda_d$, $\lambda_g$ and $\lambda_z$ during training, and the loss for joint training can be written as

$$\mathcal{L} = \lambda_d \cdot \mathcal{L}_{\text{D}} + \lambda_g \cdot \mathcal{L}_{\text{G}} + \lambda_z \cdot \mathcal{L}_{\text{z}}. \tag{13}$$

Since Weibull distributions have easy reparameterization functions, these parameters can be jointly trained by stochastic gradient descent with low-variance gradient estimation. We summarize the training algorithm at Algorithm 1 in Appendix A.4. For downstream applications, the encoder is applied to the input graph without any masking in the inference stage. The generated factor-wise node embeddings $H_d$ and graph-level embeddings $H_g$ can either be concatenated in the feature dimensions or used separately. The resulting final representation $H$ can be employed for various graph learning tasks, such as node classification and graph classification. For graph-level tasks, we use a non-parameterized graph pooling (readout) function, *e.g.*, MaxPooling and MeanPooling to obtain the graph-level representation.

**Time and space complexity:** Let's recall that in our context, $N$, $M$, and $K$ represent the number of nodes, edges, and latent factors in the graph, respectively. The feature dimension is denoted by $F$, while $L_1$, $L_2$, $L_3$, and $L_4$ represent the number of layers in the latent factor learning encoder, the latent factor-wise GMAE's encoder, the graph-level GMAE's encoder, and the decoder respectively. In DiGGR, we constrain the hidden dimension size in latent factor-wise GMAE's encoder to be $1/K$ of the typical baseline dimensions. Consequently, the time complexity for training DiGGR can be expressed as $O((L_1 + L_2 + L_3)MF + (L_1 + L_2/K + L_3)NF^2 + N^2F + L_4NF^2)$, and the space complexity is $O((L_1 + L_2 + L_3 + L_4)NF + KM + (L1 + L2/K + L3 + L4)F^2)$, with $O((L_1 + L_2/K + L_3 + L_4)F^2)$ attributed to model parameters. We utilize the Bayesian factor model in our approach to reconstruct edges. Its time complexity aligns with that of variational inference in SeeGera Li et al. [2023b], predominantly at $O(N^2F)$; Therefore, the complexity of DiGGR is comparable to previous works.

Table 1: Experiment results for node classification. Micro-F1 score is reported for PPI, and accuracy for other datasets. The best unsupervised method scores in each dataset are highlighted in bold.

| Methods | Cora | Citeseer | Pubmed | PPI |
|---|---|---|---|---|
| GCN [Kipf and Welling, 2016a] | 81.50 | 70.30 | 79.00 | $75.70 \pm 0.10$ |
| GAT [Velickovic et al., 2017] | $83.00 \pm 0.70$ | $72.50 \pm 0.70$ | $79.00 \pm 0.30$ | $97.30 \pm 0.20$ |
| DisenGCN[Ma et al., 2019] | 83.7 | 73.4 | 80.5 | - |
| VEPM[He et al., 2022b] | $84.3 \pm 0.1$ | $72.5 \pm 0.1$ | $82.4 \pm 0.2$ | - |
| MVGRL [Hassani and Khasahmadi, 2020] | $83.50 \pm 0.40$ | $73.30 \pm 0.50$ | $80.10 \pm 0.70$ | - |
| InfoGCL [Xu et al., 2021] | $83.50 \pm 0.30$ | $73.50 \pm 0.40$ | $79.10 \pm 0.20$ | - |
| DGI [Veličković et al., 2018] | $82.30 \pm 0.60$ | $71.80 \pm 0.70$ | $76.80 \pm 0.60$ | $63.80 \pm 0.20$ |
| GRACE [Zhu et al., 2020] | $81.90 \pm 0.40$ | $71.20 \pm 0.50$ | $80.60 \pm 0.40$ | $69.71 \pm 0.17$ |
| BGRL [Thakoor et al., 2021] | $82.70 \pm 0.60$ | $71.10 \pm 0.80$ | $79.60 \pm 0.50$ | $73.63 \pm 0.16$ |
| CCA-SSG [Zhang et al., 2021] | $84.20 \pm 0.40$ | $73.10 \pm 0.30$ | $81.00 \pm 0.40$ | $73.34 \pm 0.17$ |
| GAE [Kipf and Welling, 2016b] | $71.50 \pm 0.40$ | $65.80 \pm 0.40$ | $72.10 \pm 0.50$ | - |
| VGAE [Kipf and Welling, 2016b] | $76.30 \pm 0.20$ | $66.80 \pm 0.20$ | $75.80 \pm 0.40$ | - |
| Bandana [Zhao et al., 2024] | $84.62 \pm 0.37$ | $73.60 \pm 0.16$ | $\mathbf{83.53} \pm 0.51$ | - |
| GiGaMAE[Shi et al., 2023] | $84.72 \pm 0.47$ | $72.31 \pm 0.50$ | - | - |
| SEEGERA [Shi et al., 2023] | $84.30 \pm 0.40$ | $73.00 \pm 0.80$ | $80.40 \pm 0.40$ | - |
| GraphMAE [Hou et al., 2022] | $84.20 \pm 0.40$ | $73.40 \pm 0.40$ | $81.10 \pm 0.40$ | $74.50 \pm 0.29$ |
| GraphMAE2[Hou et al., 2023] | $84.50 \pm 0.60$ | $73.40 \pm 0.30$ | $81.40 \pm 0.50$ | - |
| **DiGGR** | $\mathbf{84.96} \pm 0.32$ | $\mathbf{73.98} \pm 0.27$ | $81.30 \pm 0.26$ | $\mathbf{78.30} \pm 0.71$ |

## 4 Experiments

We compare the proposed self-supervised framework DiGGR against related baselines on two fundamental tasks: unsupervised representation learning on *node classification* and *graph classification*. We evaluate DiGGR on 11 benchmarks. For node classification, we use 3 citation networks (Cora, Citeseer, Pubmed [Yang et al., 2016]), and protein-protein interaction networks (PPI) [Hamilton et al., 2017]. For graph classification, we use 3 bioinformatics datasets (MUTAG, NCI1, PROTEINS) and 4 social network datasets (IMDB-BINARY, IMDB-MULTI, REDDIT-BINARY and COLLAB). The specific information of the dataset and the hyperparameters used by the network are listed in the Appendix A.2 in table 5 and 6. We also provide the detailed experiment setup in Appendix A.2 for node classification (4.1) and graph classification (4.2)

### 4.1 Node Classification

The baseline models for node classification can be divided into three categories: $i$) supervised methods, including GCN [Kipf and Welling, 2016a] , DisenGCN[Ma et al., 2019], VEPM[He et al., 2022b] and GAT [Velickovic et al., 2017]; $ii$) contrastive learning methods, including MVGRL [Hassani and Khasahmadi, 2020], InfoGCL [Xu et al., 2021], DGI [Veličković et al., 2018], GRACE [Zhu et al., 2020], BGRL [Thakoor et al., 2021] and CCA-SSG [Zhang et al., 2021]; $iii$) generative learning methods, including GraphMAE [Hou et al., 2022], GraphMAE2[Hou et al., 2023], Bandana[Zhao et al., 2024], GiGaMAE[Shi et al., 2023], SeeGera[Li et al., 2023b], GAE and VGAE [Kipf and Welling, 2016b]. The node classification results were listed in Table 1. DiGGR demonstrates competitive results on the provided dataset, achieving results comparable to those of supervised methods.

### 4.2 Graph Classification

**Baseline Models** We categorized the baseline models into four groups: $i$) supervised methods, including GIN [Xu et al., 2018], DiffPool[Ying et al., 2018] and VEPM[He et al., 2022b]; $ii$) classical graph kernel methods: Weisfeiler-Lehman sub-tree kernel (WL) [Shervashidze et al., 2011] and deep graph kernel (DGK) [Yanardag and Vishwanathan, 2015]; $iii$) contrastive learning methods, including GCC [Qiu et al., 2020], graph2vec [Narayanan et al., 2017], Infograph [Sun et al., 2019], GraphCL [You et al., 2020], JOAO [You et al., 2021], MVGRL [Hassani and Khasahmadi, 2020], and InfoGCL [Xu et al., 2021]; $4$) generative learning methods, including graph2vec [Narayanan et al., 2017], sub2vec [Adhikari et al., 2018], node2vec [Grover and Leskovec, 2016], GraphMAE [Hou et al., 2022], GraphMAE2[Hou et al., 2023], GAE and VGAE [Kipf and Welling, 2016b]. Per graph classification research tradition, we report results from previous papers if available.

Table 2: Experiment results in unsupervised representation learning for graph classification. We report accuracy (%) for all datasets. The optimal outcomes for methods, excluding supervised approaches (GIN and DiffPool), on each dataset are emphasized in bold.

| Methods | IMDB-B | IMDB-M | MUTAG | NCI1 | REDDIT-B | PROTEINS | COLLAB |
|---------|--------|--------|-------|------|----------|----------|--------|
| GIN | 75.1± 5.1 | 52.3 ± 2.8 | 89.4 ± 5.6 | 82.7 ± 1.7 | 92.4 ± 2.5 | 76.2 ± 2.8 | 80.2 ± 1.9 |
| DiffPool | 72.6 ± 3.9 | - | 85.0 ± 10.3 | - | 92.1 ± 2.6 | 75.1 ± 3.5 | 78.9 ± 2.3 |
| VEPM | 76.7 ± 3.1 | 54.1 ± 2.1 | 93.6 ± 3.4 | 83.9 ± 1.8 | 90.5 ± 1.8 | 80.5 ± 2.8 | - |
| WL | 72.30 ± 3.44 | 46.95 ± 0.46 | 80.72 ± 3.00 | 80.31 ± 0.46 | 68.82 ± 0.41 | 72.92 ± 0.56 | - |
| DGK | 66.96 ± 0.56 | 44.55 ± 0.52 | 87.44 ± 2.72 | 80.31 ± 0.46 | 78.04 ± 0.39 | 73.30 ± 0.82 | 73.09 ± 0.25 |
| Infograph | 73.03 ± 0.87 | 49.69 ± 0.53 | 89.01 ± 1.13 | 76.20 ± 1.06 | 82.50 ± 1.42 | 74.44 ± 0.31 | 70.65 ± 1.13 |
| GraphCL | 71.14 ± 0.44 | 48.58 ± 0.67 | 86.80 ± 1.34 | 77.87 ± 0.41 | **89.53 ± 0.84** | 74.39 ± 0.45 | 71.36 ± 1.15 |
| JOAO | 70.21 ± 3.08 | 49.20 ± 0.77 | 87.35 ± 1.02 | 78.07 ± 0.47 | 85.29 ± 1.35 | 74.55 ± 0.41 | 69.50 ± 0.36 |
| GCC | 72.0 | 49.4 | - | - | 89.9 | - | 78.9 |
| MVGRL | 74.20 ± 0.70 | 51.20 ± 0.50 | 89.70 ± 1.10 | - | 84.50 ± 0.60 | - | - |
| InfoGCL | 75.10 ± 0.90 | 51.40 ± 0.80 | **91.20 ± 1.30** | 80.20 ± 0.60 | - | - | 80.00 ± 1.30 |
| graph2vec | 71.10 ± 0.54 | 50.44 ± 0.87 | 83.15 ± 9.25 | 73.22 ± 1.81 | 75.78 ± 1.03 | 73.30 ± 2.05 | - |
| sub2vec | 55.3 ± 1.5 | 36.7 ± 0.8 | 61.1 ± 15.8 | 52.8 ± 1.5 | 71.5 ± 0.4 | 53.0 ± 5.6 | - |
| node2vec | - | - | 72.6 ± 10.2 | 54.9 ± 1.6 | - | 57.5 ± 3.6 | - |
| GAE | 52.1 ± 0.2 | - | 84.0 ± 0.6 | 73.3 ± 0.6 | 74.8± 0.2 | 74.1 ± 0.5 | - |
| VGAE | 52.1 ± 0.2 | - | 84.4 ± 0.6 | 73.7 ± 0.3 | 74.8 ± 0.2 | 74.8 ± 0.2 | - |
| GraphMAE | 75.52 ± 0.66 | 51.63 ± 0.52 | 88.19 ± 1.26 | 80.40 ± 0.30 | 88.01± 0.19 | 75.30 ± 0.39 | 80.32 ± 0.46 |
| GraphMAE2 | 73.88 ± 0.53 | 51.80 ± 0.60 | 86.63 ± 1.33 | 78.56 ± 0.26 | 76.84 ± 0.21 | 74.86 ± 0.34 | 77.59 ± 0.22 |
| **DiGGR** | **77.68 ± 0.48** | **54.77 ± 2.63** | 88.72 ± 1.03 | **81.23 ± 0.40** | 88.19 ± 0.28 | **77.40 ± 0.05** | **83.76 ± 3.70** |

**Performance Comparison** The graph classification results are presented in Table 2. In general, we find that DiGGR gained the best performance among other baselines on five out of seven datasets, while achieving competitive results on the other two datasets. The performance of DiGGR is comparable to that of supervised learning methods. For instance, the accuracy on IMDB-B and IMDB-M surpasses that of GIN and DiffPool. Moreover, within the reported datasets, our method demonstrates improved performance compared to random mask methods like GraphMAE, particularly on the IMDB-M, COLLAB, and PROTEINS datasets. This underscores the effectiveness of the proposed method.

## 4.3 Exploratory Studies

**Visualizing latent representations** To examine the influence of the learned latent factor on classification results, we visualized the latent disentangled factor $z$, which reflects the node-factor affiliation, and the hidden representation $H$ used for classification. MUTAG is selected as the representative for classification benchmarks. We encodes the representations into 2-D space via t-SNE [Van der Maaten and Hinton, 2008]. The result is shown in Figure 3(a), where each node is colored according to its node labels. The clusters in Figure 3(a) still exhibit differentiation in the absence of label supervision, suggesting that $z$ obtained through unsupervised learning can enhance node information and offer a guidance for the mask modeling. We then visualize the hidden representation used for classification tasks, and color each node according to the latent factor to which it belongs. The results are depicted in Figure 3(b), showcasing separability among different color clusters. This illustrates the model's ability to extract information from the latent factor, thereby enhancing the quality of the learned representations.

**Task-relevant factors** To assess the statistical correlation between the learned latent factor and the task, we follow the approach in [He et al., 2022b] and compute the Normalized Mutual Information (NMI) between the nodes in the factor label and the actual node labels. NMI is a metric that ranges from 0 to 1, where higher values signify more robust statistical dependencies between two random variables. In the experiment, we utilized the MUTAG dataset, comprising 7 distinct node types, and the NMI value we obtained was 0.5458. These results highlight that the latent factors obtained through self-supervised training are meaningful for the task, enhancing the correlation between the inferred latent factors and the task.

**Disentangled representations** To assess DiGGR's capability to disentangle the learned representation for downstream task, we provide a qualitative evaluation by plotting the correlation of the

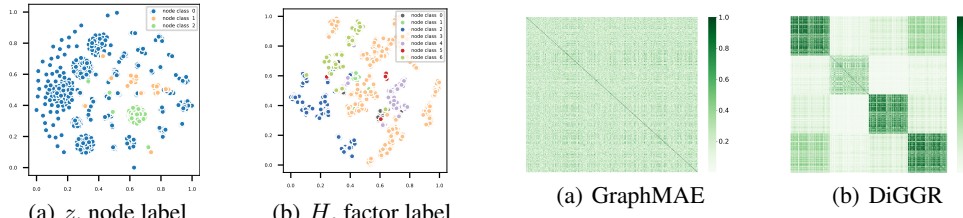

| (a) $z$, node label | (b) $H$, factor label | (a) GraphMAE | (b) DiGGR |

Figure 3: T-SNE visualization of MUTAG dataset, where $z$ is the latent factor, $H$ is the learned node representation used for downstream tasks.

Figure 4: representation correlation matrix on Cora with number of factors $K = 4$. 4(a) depicts the representation of entanglement, while 4(b) illustrates disentanglement.

Table 3: The NMI between the latent factors extracted by DiGGR and Non-probabilistic factor learning method across various datasets, and its performance improvement compared to GraphMAE, are examined. A lower NMI indicates a more pronounced disentanglement between factor-specific graphs, resulting in a greater performance enhancement.

|  | Dataset | RDT-B | MUTAG | NCI-1 | IMDB-B | PROTEINS | COLLAB | IMDB-M |
|---|---|---|---|---|---|---|---|---|
| DiGGR | NMI | 0.95 | 0.90 | 0.89 | 0.82 | 0.76 | 0.35 | 0.24 |
|  | ACC Gain | + 0.18% | + 0.53% | + 0.83% | + 2.16% | + 2.1% | + 3.44% | + 3.14% |
| Non-probabilistic Factor Learning | NMI | 1.00 | 1.00 | 0.80 | 1.00 | 0.60 | 1.00 | 0.94 |
|  | ACC Gain | -2.23% | -2.02% | -0.45% | -0.80% | -2.15% | -3.00% | -0.11% |

node representation in Figure 4. The figure shows the absolute values of the correlation between the elements of 512-dimensional graph representation and representation obtained from GraphMAE and DiGGR, respectively. From the results, we can see that the representation produced by GraphMAE exhibits entanglement, whereas DiGGR's representation displays a overall block-level pattern, indicating that DiGGR can capture mutually exclusive information in the graph and disentangle the hidden representation to some extent. Results for more datasets can be found in Appendix A.3.

**Why DiGGR works better:** To validate that disentangled learning can indeed enhance the quality of the representations learned by GMAE, we further conduct quantitative experiments. The Normalized Mutual Information (NMI) is used to quantify the disentangling degree of different datasets. Generally, the NMI represents the similarity of node sets between different factor-specific graphs, and the *lower NMI suggests a better-disentangled degree* with lower similarity among factor-specific graphs. The NMI between latent factors and the corresponding performance gain (compared to GraphMAE) are shown in the Table.3. As the results show, DiGGR's performance improvement has a positive correlation with disentangled degree, where the better the disentangled degree, the more significant the performance improvement. For methods relying on Non-probabilistic Factor Learning, the NMI tends to approach 1. This is attributed to the challenges faced by the factor learning module in converging, thereby hindering the learning of distinct latent factors. The presence of confused latent factors offers misleading guidance for representation learning, consequently leading to decreased performance.

## 5 Conclusions

In this paper, we propose DiGGR (Disentangled Generative Graph Representation Learning), designed to achieve disentangled representations in graph masked autoencoders by leveraging latent disentangled factors. In particular, we achieve this by two steps: 1) We utilize a probabilistic graph generation model to factorize the graph via the learned disentangled latent factor; 2) We develop a Disentangled Graph Masked Autoencoder framework, with the aim of integrating the disentangled information into the representation learning of Graph Masked Autoencoders. Experiments demonstrate that our model can acquire disentangled representations, and achieve favorable results on downstream tasks.

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

# A Appendix / supplemental material

Optionally include supplemental material (complete proofs, additional experiments and plots) in appendix. All such materials **SHOULD be included in the main submission.**

## A.1 Ablation Study

### A.1.1 Number of factors

One of the crucial hyperparameters in DiGGR is the *number of latent factors*, denoted as $K$. When $K = 1$ DiGGR degenerates into ordinary GMAE, only performing random masking over the entire input graph on the nodes. The influence of tuning $K$ is illustrated in Figure 5. Given the relatively small size of the graphs in the dataset, the number of meaningful latent disentangled factor $z$ is not expected to be very large. The optimal number of $z$ that maximizes performance tends to be concentrated in the range of 2-4.

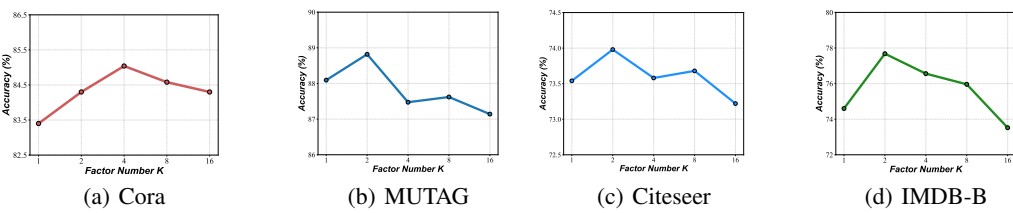

|  (a) Cora  |  (b) MUTAG  |  (c) Citeseer  |  (d) IMDB-B  |

Figure 5: Performance of the task under different choices of latent factor number $K$, where the horizontal axis represents the change in $K$ and the vertical axis is accuracy.

### A.1.2 Representation for downstream tasks

We investigate the impact of various combinations of representation levels on downstream tasks. As illustrated in Table 4, for the node classification task, both $H_d$ and $H_g$ are required, *i.e.*, concatenating them in feature dimension, whereas for the graph classification task, $H_d$ alone is sufficient. This difference may be due to the former not utilizing pooling operations, while the latter does. Specifically, the graph pooling operation aggregates information from all nodes, providing a comprehensive view of the entire graph structure. Thus, in node classification, where the node representation has not undergone pooling, a graph-level representation ($H_g$) is more critical. In contrast, in graph classification, the node representation undergoes pooling, making disentangled information $H_d$ more effective.

Table 4: The average accuracy of datasets is calculated through 5 random initialization tests when using different representations.

| $H_d$ | $H_g$ | Cora | IMDB-MULTI | Citeseer | PROTEINS |
|:---:|:---:|:---:|:---:|:---:|:---:|
| ✓ |  | $61.10 \pm 1.83$ | $\mathbf{54.77} \pm 2.63$ | $71.82 \pm 0.98$ | $\mathbf{77.76} \pm 2.46$ |
|  | ✓ | $84.22 \pm 0.38$ | $51.62 \pm 0.61$ | $73.41 \pm 0.43$ | $75.52 \pm 0.49$ |
| ✓ | ✓ | $\mathbf{84.96} \pm 0.32$ | $53.69 \pm 2.06$ | $\mathbf{73.98} \pm 0.27$ | $77.61 \pm 0.97$ |

## A.2 Implementation Details

**Environment** All experiments are conducted on Linux servers equipped with an 12th Gen Intel(R) Core(TM) i7-12700, 256GB RAM and a NVIDIA 3090 GPU. Models of node and graph classification are implemented in PyTorch version 1.12.1, scikit-learn version 1.0.2 and Python 3.7.

**Experiment Setup for Node Classification** The node classification task involves predicting the unknown node labels in networks. Cora, Citeseer, and Pubmed are employed for transductive learning, whereas PPI follows the inductive setup outlined in GraphSage [Hamilton et al., 2017]. For evaluation,

Table 5: Statistics for node classification datasets.

| | Dataset | Cora | Citeseer | Pubmed | PPI |
|---|---|---|---|---|---|
| Statistics | # node | 2708 | 3327 | 19717 | 56944 |
| | # feature | 1433 | 3703 | 500 | 50 |
| | # edges | 5429 | 4732 | 44338 | 818736 |
| | # classes | 7(s) | 6(s) | 3(s) | 121(m) |
| Hyper-parameters | Mask Rate | 0.5 | 0.5 | 0.75 | 0.5 |
| | Hidden Size | 512 | 512 | 1024 | 1024 |
| | Max Epoch | 1750 | 200 | 1000 | 1000 |
| | $\lambda_{\mathbf{d}}; \lambda_{\mathbf{g}}; \lambda_{\mathbf{z}}$ | 1; 1; 1 | 1; 1; 2 | 1; 1; 1 | 1; 1; 1 |
| | Learning Rate | 0.001 | 0.0005 | 0.001 | 0.0001 |
| | Factor_Num | 4 | 4 | 2 | 2 |

we use the concatenated representations of $H_d$ and $H_g$ in the feature dimension for the downstream task. We then train a linear classifier, report the mean accuracy on the test nodes through 5 random initializations. The graph encoder $\text{GNN}_{\text{enc}(.)}$ and decoder $\text{GNN}_{\text{dec}}(.)$ are both specified as standard GAT [Velickovic et al., 2017].We train the model using Adam Optimizer with $\beta_1 = 0.9$, $\beta_2 = 0.999$, $\epsilon = 1 \times 10^8$, and we use the cosine learning rate decay without warmup. We follow the public data splits of Cora, Citeseer, and PubMed.

**Experiment Setup for Graph Classification** The graph classification experiment was conducted on 7 benchmarks, in which node labels are used as input features in MUTAG, PROTEINS and NCI1, and node degrees are used in IMDB-BINARY, IMDB-MULTI, REDDIT-BINARY, and COLLAB. The backbone of encoder and decoder is GIN [Xu et al., 2018], which is commonly used in previous graph classification works. The evaluation protocol primarily follows GraphMAE [Hou et al., 2022]. Notice that we only utilize the factor-wise latent representation $H_d$ for the downstream task. Subsequently, we feed it into a downstream LIBSVM [Chang and Lin, 2011] classifier to predict the label and report the mean 10-fold cross-validation accuracy with standard deviation after 5 runs. We set the initial learning rate to 0.0005 with cosine learning rate decay for most cases. For the evaluation, the parameter C of SVM is searched in the sets $\{10^3, ..., 10\}$.

**Data Preparation** The node features for the citation networks (Cora, Citeseer, Pubmed) are bag-of-words document representations. For the protein-protein interaction networks (PPI), the features of each node are composed of positional gene sets, motif gene sets and immunological signatures (50 in total). For graph classification, the MUTAG, PROTEINS, and NCI1 datasets utilize node labels as node features, represented in the form of one-hot encoding. For IMDB-B, IMDB-M, REDDIT-B, and COLLAB, which lack node features, we utilize the node degree and convert it into a one-hot encoding as a substitute feature. The maximum node degree is set to 400. Nodes with degrees surpassing 400 are uniformly treated as having a degree of 400, following the methodology of GraphMAE[Hou et al., 2022]. Table 5 and Table 6 show the specific statistics of used datasets.

**Details for Visualization** MUTAG is selected as the representative benchmark for visualization in 4.3. The MUTAG dataset comprises 3,371 nodes with seven node types. The distribution is highly skewed, as 3,333 nodes belong to three types, while the remaining four types collectively represent less than 1.2% of the nodes. For clarity in legend display, we have visualized only the nodes belonging to the first three types.

### A.3 Disentangled Representations Visualization

We chose PROTEINS and IMDB-MULTI as representatives of the graph classification dataset, and followed the same methodology as in Section 4.3 to visualize their representation correlation matrices on GraphMAE, and community representation correlation matrices on DiGGR, respectively. The feature dimensions of PROTEINS and IMDB-MULTI are both 512 dimensions, and the number of communities is set to 4.

Table 6: Statistics for graph classification datasets.

| | Dataset | IMDB-B | IMDB-M | PROTEINS | COLLAB | MUTAG | REDDIT-B | NCI1 |
|---|---|---|---|---|---|---|---|---|
| Statistics | Avg. # node | 19.8 | 13.0 | 39.1 | 74.5 | 17.9 | 429.7 | 29.8 |
| | # features | 136 | 89 | 3 | 401 | 7 | 401 | 37 |
| | # graphs | 1000 | 1500 | 1113 | 5000 | 188 | 2000 | 4110 |
| | # classes | 2 | 3 | 2 | 3 | 2 | 2 | 2 |
| Hyper-parameters | Mask Rate | 0.5 | 0.5 | 0.5 | 0.75 | 0.75 | 0.75 | 0.25 |
| | Hidden Size | 512 | 512 | 512 | 256 | 32 | 512 | 1024 |
| | Max Epoch | 300 | 200 | 50 | 20 | 20 | 200 | 200 |
| | Learning Rate | 0.0001 | 0.001 | 0.0005 | 0.001 | 0.001 | 0.0005 | 0.0005 |
| | $\lambda_{\mathbf{d}}; \lambda_{\mathbf{g}}; \lambda_{\mathbf{z}}$ | 1; 1; 1 | 1; 1; 1 | 1; 1; 1 | 1; 1; 1 | 1; 1; 1 | 1; 1; 1 | 1; 0.5; 1 |
| | Batch_Size | 32 | 32 | 32 | 32 | 32 | 16 | 32 |
| | Pooling_Type | mean | mean | max | max | sum | max | max |
| | Factor_Num | 2 | 4 | 4 | 4 | 2 | 2 | 4 |

The result is presented in Figure 6. We can see from the results that the graph representations of GraphMAE are entangled. In contrast, the correlation pattern exhibited by DiGGR reveals four distinct diagonal blocks. This suggests that DiGGR is proficient at capturing mutually exclusive information within the latent factor, resulting in disentangled representations.

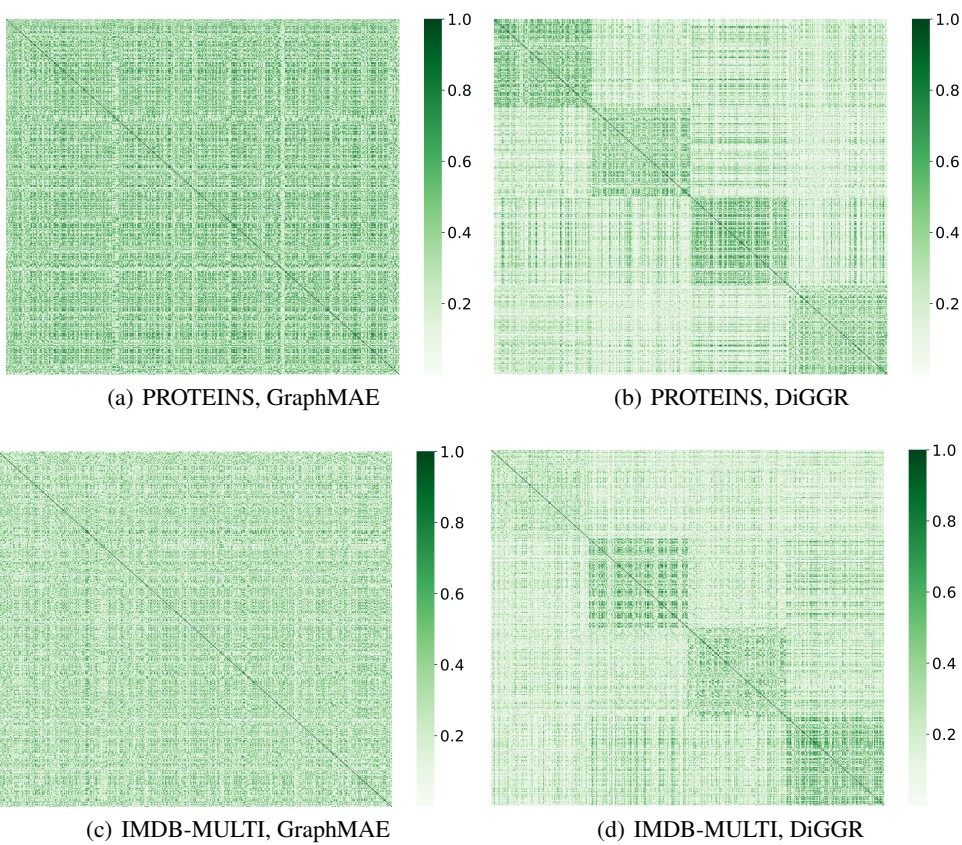

(a) PROTEINS, GraphMAE

(b) PROTEINS, DiGGR

(c) IMDB-MULTI, GraphMAE

(d) IMDB-MULTI, DiGGR

Figure 6: The absolute correlation between the representations learned by GraphMAE and DiGGR is measured on the **PROTEINS** and **IMDB-MULTI** datasets when $K = 4$.

## A.4 Training Algorithm

**Algorithm 1** The Overall Training Algorithm of DiGGR
---
1: **Input**: Graph $\mathcal{G} = \{V, A, X\}$; latent factor number $K$.

2: **Parameters**: $\Theta$ in the inference network of Latent Factor Learning phase, $\boldsymbol{\Omega}$ in the encoding network of DiGGR, $\boldsymbol{\Psi}$ in the decoding network of DiGGR.

3: Initialize $\Theta$, $\boldsymbol{\Omega}$, and $\boldsymbol{\Psi}$;

4: **for** iter = 1,2, $\cdots$ **do**

5:     Infer the *variational posterior* of $\mathbf{z_u}$ based on Eq. 5;

6:     Sample latent factors $\mathbf{z_u}$ from the variational posterior according to Eq. 6;

7:     Factorize the graph $\mathcal{G}$ into $K$ factor-wise groups $\{\mathcal{G}^{(\mathbf{k})}\}_{\mathbf{k=1}}^{\mathbf{K}}$ by node and edge factorization methods;

8:     Encoding $\{\mathcal{G}^{(\mathbf{k})}\}_{\mathbf{k=1}}^{\mathbf{K}}$ via latent factor-wise Graph Masked Autoencoder according to Eq. 8;

9:     Encoding $\mathcal{G}$ via graph-level graph masked autoencoder according to Eq. 11;

10:     Calculate $\nabla_{\Theta, \boldsymbol{\Omega}, \boldsymbol{\Psi}} \mathcal{L}(\Theta, \boldsymbol{\Omega}, \boldsymbol{\Psi}; \mathcal{G})$ according to Eq. 13, and update parameters $\Theta$, $\boldsymbol{\Omega}$, and $\boldsymbol{\Psi}$ jointly.

11: **end for**=0
---

## A.5 Broader Impacts

This paper presents work whose goal is to advance the field of Machine Learning. There are many potential societal consequences of our work, none which we feel must be specifically highlighted here.

## A.6 Limitations

Despite the promising experimental justifications, our work might potentially suffer from limitation: Although the complexity of the model is discussed in Section 3.4, and it is comparable to previously published work, extending DiGGR to extremely large graph datasets remains challenging at this stage due to the incorporation of an additional probabilistic model into the generative graph framework. One potential solution to this problem could be utilizing PPR-Nibble [Andersen et al., 2006] for efficient implementation, a method that has proven effective in some graph generative models [Hou et al., 2023]. This approach will be pursued in our future work.

