# OpenReview forum: "Disentangled Generative Graph Representation Learning"
_NeurIPS.cc/2024/Conference — Submitted to NeurIPS 2024_

### Official Review · Reviewer_efGr · 2024-07-09

**Soundness:** 3
**Presentation:** 2
**Contribution:** 3
**Rating:** 4
**Confidence:** 3

**Summary:**

The paper introduces DiGGR (Disentangled Generative Graph Representation Learning), a self-supervised learning framework that aims to guide graph mask modeling through disentangled latent factors to enhance the disentanglement of learned representations. Extensive experiments across 11 public datasets for node and graph classification tasks demonstrate the framework's effectiveness, significantly outperforming many existing self-supervised methods.

**Strengths:**

Innovative Approach: The DiGGR framework innovatively utilizes disentangled latent factors to guide graph mask modeling, a novel contribution in generative graph representation learning that significantly enhances the model's explainability and robustness.
Comprehensive Experiments: The paper conducts extensive experiments on multiple datasets and tasks, showing significant performance improvements over existing methods, thus providing strong empirical support for the proposed approach.

**Weaknesses:**

Complexity and Scalability: The framework appears computationally complex, which might limit its scalability to very large graphs or real-time applications. Unfortunately, this aspect is not extensively discussed in the paper.
Lack of Theoretical Analysis: While the empirical results are strong, the paper lacks a detailed theoretical analysis of why the disentanglement process improves performance, which could provide deeper insights into the method’s efficacy and limitations.

**Questions:**

See Weaknesses.

**Limitations:**

See Weaknesses.

---

> ### Author Rebuttal · Authors · 2024-08-05
>
> ## Q1: Computation Comleaxity
>
> Grateful for your comments. We will expand the discussion on complexity and scalability in our revisions, focusing on the following three aspects:
>
> 1. Complexity Analysis: We discussed the network's complexity in Section 3.4 and will later compare the time and space complexity of additional models to demonstrate that DiGGR is comparable to previous work in terms of complexity.
>
> 2. Empirical Results: we compared DiGGR and SeeGera [1] on the Cora dataset. All experiments were conducted on an RTX 3090, with all the models adhering to their default settings, and perform a single run with the same random seed and the same Pytorch platform.
>
> |   Method  | Training Time |  epoch / second  |  Model Size | Training GPU Memory |
> |:---------:|:-------------:|:----------------:|:-----------:|:-------------------:|
> |  SeeGera  |   200 second  |    5.28 it /s    |    25 MB    |      588.02 MB      |
> | **DiGGR** | **59 second** | **29.35 it / s** | **23.2 MB** |    **566.83 MB**    |
>
> The experimental results indicate that our model has lower training time and resource consumption compared to SeeGera. Specifically, the training time of DiGGR is nearly four times less than that of SeeGera. These results collectively demonstrate that DiGGR maintains a manageable overall complexity while achieving excellent performance.
>
> 3. Lastly, we are actively working on extending DiGGR with a focus on scaling our algorithm to extremely large graphs. We conducted additional experiments on the larger dataset ogbn-arxiv. During the reconstruction of the adjacency matrix, we used the method from [5], sampling only a portion of the matrix for computation each time. The results are as follows:
>
> |     Methods    |    ogbn-arxiv    |
> |:-------------------:|:----------------:|
> | GraphMAE  [2] |   71.75 ± 0.17   |
> | GraphMAE2 [3] |   71.95 ± 0.08   |
> |     DiGGR     | **72.12 ± 0.08** |
>
> DiGGR outperforms both on the ogbn-arxiv dataset, validating our approach's effectiveness and scalability. Previous works like GraphMAE initially focused on smaller datasets, but its subsequent work, GraphMAE2, used the PPR-Nibble technique to extend to large graph data. We believe that DiGGR also has the potential for scaling to large datasets and are actively working on advancing this capability.
>
>
> ## Q2: Theoretical Analysis
>
> Thank you for your valuable suggestion. We would like to provide a detailed theoretical analysis as follows:
>
> DiGGR is built on an graph autoencoder (GAE)-based framework. Recent studies [6, 7] have demonstrated a direct connection between GAE and contrastive learning through a specific form of the objective function. The loss function can be rewritten as follows:
> $$
> L^+ = \frac{1}{|\varepsilon^+|}\sum_{(u, v) \in V^+}{\log f_{dec}(h_u, h_v)}
> $$
> $$
> L^- = \frac{1}{|\varepsilon^-|}\sum_{(u', v') \in V^-}{\log (1 - f_{dec}(h_{u'}, h_{v'}))}
> $$
> $$
> L_{GAE} = -(L^+ + L^-)
> $$
> where $h_u$ and $h_v$ are the node representations of node $u$ and node $v$ obtained from an encoder $f_{enc}$ respectively (eg., a GNN); $\varepsilon^+$ is a set of positive edges while $\varepsilon^-$ is a set of negative edges sampled from graph, and $f_{dec}$ is a decoder; Typically, $\varepsilon^+ = \varepsilon$.
>
> Building on recent advances in information-theoretic approaches to contrastive learning [8, 9], a recent study [6] suggests that for SSL pretraining to succeed in downstream tasks, task-irrelevant information must be reasonably controlled. Therefore, the following proposition is put forward:
>
> The task irrelevant information $I(U; V| T)$ of GAE can be lower bounded with:
> $$
> I(U; V| T) \geq \frac{(E[N_{uv}^{k}])^2}{N_k} . \gamma^2
> $$
> minimizing the aforementioned $L_{GAE}$ is in population equivalent to maximizing the mutual information between the k-hop subgraphs of adjacent nodes, and **the redundancy of GAE scales almost linearly with the size of overlapping subgraphs**.
>
> The above proposition has been proved in detailed in [6], where $I(.;.)$ is the mutual information, $U$ and $V$ be random variables of the two contrasting views, and $T$ denote the target of the downstream task. $N_{uv}^{k}$ is the size of the overlapping subgraph of $G^k(u)$ and $G^k(v)$, and the expectation is taken with respect to the generating distribution of the graph and the randomness in choosing $u$ and $v$.
>
> According to this lower bound, we need to reduce the task-irrelevant redundancy to design a better graph SSL methods. In DiGGR, we first factorize the input graph based on latent factor learning before feeding it into the masked autoencoder. Take **Figure 1 from the PDF in global rebuttal** as an example. Nodes $a$ and $b$ have overlapping 1-hop subgraphs. However, after graph factorization, the connection between $a$ and $b$ is severed, thereby reducing large-scale subgraph overlap and lowering the lower bound of task-irrelevant information. As shown in Table 3 of the paper, after factorization, the latent factor groups extracted by DiGGR exhibit lower normalized mutual information (NMI), indicating reduced overlap between the latent factor groups. This result aligns with our theoretical analysis and highlights the advantages of our proposed method.
>
> [1] Seegera: Self-supervised semi-implicit graph variational auto-encoders with masking. WWW 2023.
>
> [2] Graphmae: Self-supervised masked graph autoencoders. KDD 2022
>
> [3] GraphMAE2: A Decoding-Enhanced Masked Self-Supervised Graph Learner. WWW 2023
>
> [4] Local graph partitioning using pagerank vectors.  (FOCS'06). IEEE, 2006
>
> [5] Fastgae: Scalable graph autoencoders with stochastic subgraph decoding.  Neural Networks 142 (2021): 1-19
>
> [6] What’s Behind the Mask: Understanding Masked Graph Modeling for Graph Autoencoders. KDD 2023
>
> [7] How Mask Matters: Towards Theoretical Understandings of Masked Autoencoders. NeurIPS 2022
>
> [8] What Makes for Good Views for Contrastive Learning?.  NeurIPS 2020
>
> [9] Self-supervised Learning from a Multi-view Perspective. ICLR 2021

---

### Official Review · Reviewer_DL9g · 2024-07-10

**Soundness:** 3
**Presentation:** 3
**Contribution:** 2
**Rating:** 4
**Confidence:** 4

**Summary:**

The paper introduces a framework called DiGGR, aimed at improving the robustness and explainability of generative graph models by addressing the issue of entangled graph representations.

**Strengths:**

1. The paper tells the story in an easy-to-read way, and the whole paper is quite easy to follow.
2. The problem of disentangled learning is a very popular yet important task.
3. The paper conducts comprehensive experiments to evaluate their method.

**Weaknesses:**

1. Lack of novelty. Graph disentangled learning is not a new task. There are tons of existing methods for disentangled representation learning, such as those maximizing KL divergence or minimizing mutual information between two sets of representations. A lot of related works such as [1], [2], [3] and [4] are not discussed. Also node factorization is not a new idea, such as node clustering in [3].

[1] Disentangled graph collaborative filtering. SIGIR 2020.
[2] Disentangled Graph Convolutional Networks. ICML 2019.
[3] Deep Generative Model for Periodic Graphs. NeurIPS 2022.
[4] Disentangled contrastive learning on graphs. NeurIPS 2021.

2. The motivation of the proposed method is not clear to me. For example, why should we use mask? Also, why the proposed method sticks to GAE, not VGAE or other types of GNN, such as GCN, GIN or GAT?

**Questions:**

1. Line 21: Does "other modalities" mean the GMAE it self, or another different kind of graph? What's the difference between GMAE and other famouse GNN such as GCN, GIN, GAT, etc?
2. Line 37, Why should we disentangle different pieces of information of a single node? Will this help improve GNN's expressiveness?
3. Line 120: The fulle name of EPM should be introduced before using its abbreviation.
4. The paper mentions "mask strategies" for quite a few times, but I'm still confused about the purpose of masking. The paper just mentions that the proposed approach uses masking but does not tell readers the intuition behind it. Will masking produce more expressive graph representation? Will it help disentanglement?
5. What's the difference between the proposed approach and a lot of existing approachs such as [1], [2], [3] and [4]? It seems that the proposed one factorizes nodes and edges and finally pull them together, while other techniques may rely on KL divergence or mutual information. I don't think those works are well discussed by the paper.

[1] Disentangled graph collaborative filtering. SIGIR 2020.
[2] Disentangled Graph Convolutional Networks. ICML 2019.
[3] Deep Generative Model for Periodic Graphs. NeurIPS 2022.
[4] Disentangled contrastive learning on graphs. NeurIPS 2021.

6. I understand that the number of latent groups can be pre-defined, but can we theoretically guarantee that those groups are identifiable in terms of the learned representation? As shown in Figure1, are those groups permutational invariant for their positions?

**Limitations:**

Yes. Limitations have been discussed in the paper.

---

> ### Author Rebuttal · Authors · 2024-08-05
>
> ### W1: Novelty
> Thank you for the time and effort you have dedicated to our paper. It is true that DiGGR and [1-4] use some common techniques in disentangled learning. However, we sitll believe our approach offers novelties in the following aspects:
>
> (i)The main goal of our paper is to use disentangled learning to enhance self-supervised feature learning, rather than focusing solely on disentanglement itself. To the best of our knowledge, most current work on graph masked autoencoders (GMAE) often overlooks the disentanglement of representations in the context of node classification and graph classification tasks: When performing mask modeling, these approaches treat the entire graph as a whole and often overlook its underlying structure.
> For example, a node $n$ might belong to both community $A$ and community $B$, playing different roles in each. Consequently, different learning strategies should be adopted for these roles. However, most GMAE methods ignore these latent relationships and uniformly sample nodes from the entire graph for masking, disregarding the heterogeneous factors present within nodes. Therefore, we introduced disentanglement into GMAE and used a tailored probabilistic approach to improve the learned representations.
>
> (ii) our method differs significantly from those in [1-4]. Please refer to the **global rebuttal**. Briefly, the main differences are as follows:
> (a) Our approach models the latent factor $z$ as a probability distribution rather than a variable. This differs from most prototype-based factor learning methods [1, 2, 4]. By modeling $z$ as a distribution, our method introduces randomness during network training, leading to faster convergence and more distinctive latent factors.
> (b) We model the latent factor using a Gamma distribution, leveraging its non-negative property to transform the factorization problem into a non-negative matrix factorization problem. This contrasts with [3], where the distribution nature prevents such a conversion. According to [5], this approach results in representations that are more disentangled and expressive.
>
> ### W2：Why not other framework
> What we provide in this article is a general framework method. Within the DiGGR framework, any other GNN, such as GCN, GIN, and GAT, can be used for representation learning. Additionally:
>
> 1. In graph SSL, contrastive methods dominate node and graph classification but face challenges: (a) reliance on manually constructed negative samples, (b) need for high-quality data augmentation. Graph autoencoders (GAEs) avoid these issues by directly reconstructing the input graph. Therefore, our work is based on a GAE framework.
>
> 2. Due to an overemphasis on structural information, GAEs have lagged behind contrastive learning in node and graph classification tasks. Recently, [6] demonstrated that masked autoencoders achieve performance comparable to contrastive learning in graph SSL, with [7,8] confirming this. Thus, we incorporated mask modeling into our network.
>
> However, Graph mask modeling is not our main focus. Our goal is to use disentangling to improve self-supervised representation learning, supported by a tailored probabilistic latent factor learning method.
>
> ### Questions
> **Q1**: 1. To clarify, "other modalities" refers to graph domain. Line 21 means While MAE techniques are well-established in language and image domains, they are now gaining attention in graph learning; 2. GMAE and GNN are analogous to MAE and Transformers. GMAE employs an encoder-decoder architecture to reconstruct input graphs, it can use GNNs as either encoders or decoders.
>
> **Q2**: Disentangling node latent structures has proven effective in graph representation learning  [1,2]. The node latent structure sets the aspects of node information. For example, In a social network, node $n$ might represent both a student and a club member, showing different traits in each context. Thus, the ideal approach is to learn distinct node properties for each group and aggregate these to represent the node’s overall information. Such an approach has been proven effective in GNNs. For example, [3] address the issue of latent factor entanglement and show that disentangled node representations, achieved through iterative neighborhood segmentation, are effective.
>
> **Q3**: We will include the full name of EPM, which is Edge Partition Model in the revision.
>
> **Q4**: We use mask strategies to boost the learning capability of the GAE framework. Please see our response in W2 for details.
>
> **Q5**: We have thoroughly reviewed [1-4]. Please refer to our detailed response in the **global rebuttal** section.
>
> **Q6**: Theoretical guarantees for identifying latent groups pose challenges for both our study and graph learning filed at large. Our algorithm encourages latent group discovery only if it improves the explanation of the defined loss terms. However, it does not guarantee identifying groups that may not be relevant to specific tasks. Thus, while it identifies meaningful groups, it may overlook less impactful ones.
> Our algorithm does not enforce permutation invariance for the discovered latent groups. Instead, it relies on the data to determine whether such invariance is necessary for the task. This approach allows for a data-driven determination of the relevance and utility of permutation invariance in the latent groups identified by our model.
>
> [1] Disentangled graph collaborative filtering. SIGIR 2020
>
> [2] Disentangled Graph Convolutional Networks. ICML 2019
>
> [3] Deep Generative Model for Periodic Graphs. NeurIPS 2022
>
> [4] Disentangled contrastive learning on graphs. NeurIPS 2021
>
> [5] Learning the parts of objects by non-negative matrix factorization. Nature 1999.
>
> [6] Graphmae: Self-supervised masked graph autoencoders. KDD 2022.
>
> [7] Seegera: Self-supervised semi-implicit graph variational auto-encoders with masking. WWW 2023.
>
> [8] Gigamae: Generalizable graph masked autoencoder via collaborative latent space reconstruction. CIKM 2023.

---

> > ### Comment · Reviewer_DL9g · 2024-08-12
> > **reviewer response**
> >
> > Thank the author or the time on rebuttal. I think they partially addressed my concern. As a result, I'm willing to increase my score. Here are the remaining questions:
> >
> > 1. I don't the treating $z$ as a distribution has any difference from existing works that rely on VAE, such as [1] and [2], since VAE also treats latents as random varibles that are supposed to follow Gaussian distribution. Basically, latents are sampled from Gaussian in VAE.
> >
> > 2. I don't see obvious benefits of modeling latents using Gamma distribution, as even for regular matrix that has values that can be either negative or positive, we can still perform matrix factorization, such as SVM. I would suggest author to clearly explain this in the paper. Also, I think with Gamma distribution assumption, e.g. the last layer of encoder, need to produce non-negative values then. Will that cause any instability during training? Also, it would be great if author could explain the benefits brought by the sparsity of Gamma distribution. Will this cause loss of information? Why not just use L1 norm to enforce the sparsity?
> >
> > 3. For Q2, are there mathematical metrics to measure the expressiveness of GNN? If so, I believe involving those metrics in experiments would be more persuasive.
> >
> > 4. Ignoring permutation invariance will introduce huge complexity for model parameters to well capture all kinds of graphs. If there are $n$ nodes of a graph, I guess it's $O(n!)$ to permute those nodes for a graph generation task.
> >
> > [1] Deep Generative Model for Periodic Graphs. NeurIPS 2022.
> >
> > [2] Multi-objective Deep Data Generation with Correlated Property Control. NeurIPS 2022.

---

> > > ### Author Response · Authors · 2024-08-12
> > > **We sincerely appreciate your efforts in reviewing the manuscript and hope our response will address your concerns effectively.**
> > >
> > > ### **Q1 and Q2**:
> > > Thank you for your valuable advice. We will incorporate the benefits of using the Gamma distribution in the next revision. Essentially, we model $z$ as a Gamma distribution rather than a Gaussian distribution for the following reasons:
> > >
> > > 1. Compared to the Gaussian distribution, the Gamma distribution possesses **non-negative** and **sparse** properties, both of which we leverage in our approach.
> > >
> > > a) Based on the non-negative characteristic, the latent factor learning can be converted into **non-negative** matrix factorization (NMF). Unlike canonical matrix factorization, NMF ensures that different features will not cancel one another out when calculating feature similarity [4]. To boot, the non-negativity constraint will lead to sort of sparseness naturally, which is proved to be a highly effective representation distinguished from both the completely distributed and the solely active component description [5]. According to [6], such property inherently benefits the learning of disentangled representation, which is unattainable with canonical matrix factorization under a Gaussian distribution.
> > >
> > > b) The sparsity induced by the Gamma distribution is akin to imposing an additional sparsity constraint on NMF. This constraint aids in enhancing the uniqueness of the decomposition while enforcing a locality-based representation, making the factorization sparser and improving its robustness against potential offsets caused by additive noise [6].
> > >
> > >
> > > 2. **Training Stability of Gamma**: We adopted the variational inference method proposed in [7], using the Weibull distribution to approximate the Gamma distribution for reparameterization. This method has been shown to enable stable training in graph networks [8]. We want to emphasize that the sparsity introduced by the Gamma distribution does not compromise the model's performance. On the contrary, recent studies [4, 5, 7] have shown that the increased feature sparsity enhances the network's nonlinearity, leading to improved model expressiveness and generalization ability. (Note that non-negative values are common in deep learning models, particularly as outputs from non-linear functions like ReLU.  These non-linear functions can enhance the model's expressive power and generalization ability by increasing its nonlinearity, rather than bringing information loss.)
> > >
> > > 3. **Why Choose Gamma instead of L1**:
> > > Gamma distribution introduces probabilistic sparsity, which has the following two benefits :
> > >
> > > a)Sparsity can be controlled more flexibly by adjusting parameters such as the shape and scale of the Gamma distribution[9]. Unlike the L1 norm, which forces some parameters to zero, this method allows for a balanced trade-off between sparsity and information retention[10], reducing the risk of information loss due to excessive sparsification.
> > >
> > > b)The optimization process of Gamma's probabilistic sparsity is smoother than that of the L1 norm[12], which causes non-smoothness and issues like "dead neurons" as parameters approach zero[11]. The sparsity introduced by Gamma can improve convergence and reduce the risk of local minima or unstable solutions[9].
> > >
> > > c) Probabilistic sparsity can provide nonlinearity, analogous to a Relu activation function, which is often helpful for the model's expressiveness. However, the L1 norm often hurts the original model's expressiveness.
> > >
> > > ### **Q3**
> > > Similar to other works [1, 2, 8], we evaluate the expressiveness of our framework using various downstream task metrics. In Tables 1 and 2 of this paper, we use accuracy for comparison. In the global response, we specifically use the Gamma latent variable model for classification and compare it with the Gaussian latent variable model from [1] on downstream tasks. These comparisons demonstrate the expressiveness of the gamma latent variable model.  In future revisions, we will also include experimental results on the link prediction task to more comprehensively validate the gamma latent variable model's expressiveness.
> > >
> > >
> > > | Model   | IMDB-BINARY (%) | MUTAG (%)   |
> > > |---------|-----------------|-------------|
> > > | PGD-VAE | 54.37 ± 0.21     | 84.06 ± 0.52|
> > > |  Gamma latent variable model | 70.79 ± 0.34     | 85.12 ± 0.64|

---

> > > > ### Author Response · Authors · 2024-08-12
> > > > **Further reply**
> > > >
> > > > ### **Q4**:
> > > > This is an insightful question, and we would like to explain it from the following two perspectives:
> > > >
> > > > 1. We have not overlooked permutation invariance in the task. Rather than explicitly enforcing it in the model, we take a data-driven approach to determine permutation invariance within the latent groups identified by DiGGR. Specifically, in our model, we define the log-likelihood of the adjacency matrix as the optimization function (see Equation 7).
> > > > According to the theorem in [3], the gradient of the log-likelihood estimation, $\nabla_A \log p(A)$, is permutation equivariant, meaning that the log-likelihood function $\log p(A)$ is inherently permutation invariant. By optimizing latent factor learning using the ELBO of $\log p(A)$, we implicitly enforce permutation invariance in our latent factor.
> > > >
> > > > 2. Our model is primarily designed for classification tasks, where no operations in the network involve permutation, and all the adjacency matrices stick to a specific and intrinsic node ordering.
> > > > If the model is extended to graph generation tasks in the future, we can adopt the approach in [1], which uses a fixed permutation to constrain the adjacency matrix A to a specific node ordering, thereby limiting the model's space complexity to $Q(n^2)$.
> > > >
> > > > [1] Deep Generative Model for Periodic Graphs. NeurIPS 2022.
> > > >
> > > > [2] Multi-objective Deep Data Generation with Correlated Property Control. NeurIPS 2022.
> > > >
> > > > [3] Permutation Invariant Graph Generation via Score-Based Generative Modeling. AISTATS 2020.
> > > >
> > > > [4] NON-NEGATIVE CONTRASTIVE LEARNING. ICLR 2024.
> > > >
> > > > [5] Nonnegative Matrix Factorization: A Comprehensive Review. TKDE 2013.
> > > >
> > > > [6] Lee, Daniel D., and H. Sebastian Seung. "Learning the parts of objects by non-negative matrix factorization." nature 401.6755 (1999): 788-791.
> > > >
> > > > [7] WHAI: WEIBULL HYBRID AUTOENCODING INFERENCE FOR DEEP TOPIC MODELING. ICLR 2018
> > > >
> > > > [8] A variational edge partition model for supervised graph representation learning. NeurIPS 2022
> > > >
> > > > [9] Kingma, Diederik P., and Max Welling. "Auto-encoding variational bayes." arXiv preprint arXiv:1312.6114 (2013).
> > > >
> > > > [10] Latent dirichlet allocation. NeurIPS 2001
> > > >
> > > > [11] Regression shrinkage and selection via the lasso. Journal of the Royal Statistical Society Series B: Statistical Methodology 58.1 (1996): 267-288.
> > > >
> > > > [12] Learning structured output representation using deep conditional generative models. NeurIPS 2015

---

> > > > > ### Author Response · Authors · 2024-08-14
> > > > >
> > > > > Dear Reviewer DL9g,
> > > > >
> > > > > I hope this email finds you well. I wanted to follow up on the responses I provided two days ago to the remaining questions and concerns you raised regarding our submission .
> > > > >
> > > > > As the deadline is approaching, I wanted to kindly ask if the clarifications have addressed your concerns. If there are any outstanding issues, please let me know, and I would be happy to address them as quickly as possible.
> > > > >
> > > > > If the responses were satisfactory, I would greatly appreciate it if you could reconsider the evaluation of our submission.
> > > > >
> > > > > Thank you again for your time and thoughtful feedback.
> > > > >
> > > > > Best regards,
> > > > >
> > > > > Paper 17865 Authors

---

### Official Review · Reviewer_Vvqv · 2024-07-14

**Soundness:** 3
**Presentation:** 3
**Contribution:** 3
**Rating:** 6
**Confidence:** 3

**Summary:**

The work proposes a disentangled generative self-supervised learning method for graphs. The authors introduce a latent factor learning module to capture the heterogeneous factors in the nodes. The proposed method factorizes the graph into factor-specific subgraphs, and jointly trains a disentangled Graph MAE applying distinct masks for each subgraph. Experimental results demonstrate that DiGGR outperforms traditional methods that treat the graph holistically, without accounting for its latent structure.

**Strengths:**

1. The proposed method first explores a factorization method for generative graph SSL.
2. The authors provide extensive experimental results and analysis on both node and graph-level tasks to show the improved effectiveness, interpretability, and generalization by using the proposed method.

**Weaknesses:**

- The computation complexity of the proposed method is quite high. Could the author pride training time comparison to the baseline methods to help us get a sense of the real complexity?
- Could the author provide more insights on how to find an optimal factor number K according to the statistics of diverse datasets? This might be useful for real-world applications.

**Questions:**

Please see the weakness part above.

**Limitations:**

Yes, the authors discussed the limitations.

---

> ### Author Rebuttal · Authors · 2024-08-05
>
> ## Q1: Computation Complexity
>
> We genuinely appreciate the time and effort you dedicated to a thorough reading of our paper. Based on your suggestions, we conducted a training time comparison experiment. We found it to be very helpful for readers to understand the actual model complexity, and we will include this experiment in our paper in the later revision.
> We have provided the time required for training on IMDB-BINAR, IMDB-MULTI and Cora in the following tables.  All experiments were conducted on an RTX 3090, with all the models adhering to their default settings, and perform a single run with the same random seed and the same Pytorch platform.
> In Section 3.4 of the paper, we analyze the time complexity of DiGGR and find it consistent with Seegera [1]. Therefore, we included it in the comparison. Since the public code of SeeGera only implement on Cora dataset, we also compared the training time of VEPM [2] on graph classification data for a comprehensive comparision.
>
>
>
> | Method  | Training Time | epoch / second |
> |---------|---------------|----------------|
> | SeeGera | 200 second     | 5.28 it /s     |
> | DiGGR   | **59 second**     | **29.35 it / s**   |
>
>
> |       | IMDB-BINARY    |                 | IMDB-MULTI     |                 |
> |-------|----------------|-----------------|----------------|-----------------|
> |       | Training Time  | second / epoch  | Training Time  | second / epoch  |
> | VEPM  | 363 second     | 1.21s/epoch     | 314 second     | 1.57 s/epoch    |
> | DiGGR | **340 second** | **1.13s/epoch** | **290 second** | **1.45s/epoch** |
>
>
> From the table, it is evident that DiGGR's training time is significantly shorter than SeeGera's during actual training. On the Cora dataset, DiGGR's overall training time is almost 4 times less than SeeGera's, and the average number of epochs per second is 5.56 times higher than SeeGera. For graph classification, DiGGR's training speed is generally faster than VEPM, with shorter runtime per epoch.
> Overall, these experimental results revalidate the conclusion in Section 3.4 of the paper: the complexity of DiGGR is comparable to previous works.
>
> [1] Li, Xiang, et al. "Seegera: Self-supervised semi-implicit graph variational auto-encoders with masking." Proceedings of the ACM web conference 2023.
>
> [2] He, Yilin, et al. "A variational edge partition model for supervised graph representation learning." Advances in Neural Information Processing Systems 35 (2022): 12339-12351.
>
>
>
> ## Q2: Optimal factor number K
>
> Thank you for your valuable comments. As you pointed out, the hyperparameter $K$ is indeed a crucial factor in our model. We would like to share our view on this parameter from two aspects:
>
> (1) Empirical Analysis: We initially analyzed the impact of different $K$ values on the datasets. We conducted ablation studies on $K$ across four different datasets (see Figure 5 in Appendix A) and documented the optimal hyperparameter $K$ used for each dataset (see Tables 5 and 6 in Appendix A). It was observed that the optimal $K$ for most datasets typically falls within the range of 2 to 4. Therefore, we believe the optimal number of latent factors depends on the dataset's complexity. In practical applications, we use NMI to measure the similarity between latent factors. If the NMI between a new latent factor and the existing ones is high, we opt to retain the original number of latent factors.
>
> (2) Performance Stability: Although the optimal $K$ for achieving the best performance may vary across different datasets, the performance variation due to different $K$ values is not significant in practical applications. For instance, in the ablation study of $K$, the performance fluctuation remained relatively small as $K$ varied from 1 to 16. On the MUTAG dataset, the standard deviation of accuracy across different $K$ values was only 0.58, and on Cora, this value was just 0.25.
>
> In summary, the optimal number of latent factors depends on the dataset's complexity. While the optimal $K$ may differ across datasets, the resulting performance variation is minimal.

---

### Official Review · Reviewer_wD15 · 2024-07-15

**Soundness:** 3
**Presentation:** 3
**Contribution:** 3
**Rating:** 4
**Confidence:** 4

**Summary:**

The paper proposes a self-supervised learning framework DiGGR, aimed at enhancing the disentanglement of learned graph representations. The authors argue that existing generative graph models tend to overlook the entanglement of learned representations, leading to non-robust and non-explainable models. DiGGR addresses this by introducing a latent factor learning module and a disentangled graph masked autoencoder, allowing for factor-wise graph representations. The framework is tested on various benchmarks, demonstrating its effectiveness in outperforming previous self-supervised methods.

**Strengths:**

1. This paper studies an interesting research problem that is disentangled graph representation learning. This research problem is very hot recently.

2. The model design is easy to understand. The paper provides a detailed explanation of the proposed model.

3. The experiments demonstrate the effectiveness of the model. The performance improvement on some comparisons seems to be significant.

**Weaknesses:**

1. One of my concerns is from the novelty. I think the model design is a little similar to the works [1-2]. The authors should make more comprehensive discussions to show the differences between them.

2. The experiments ignore some recent or related contrastive baselines [1-4] for comparisons.  The improvements on some datasets seem to be not significant.

3.  More large-scale benchmarks should also be considered, e.g., OGB. The experimental settings are not very clear for reproducing the results.

[1] Disentangled contrastive learning on graphs. NeurIPS 2021.

[2] Disentangled Graph Contrastive Learning With Independence Promotion. TKDE 2022.

[3] Augmentation-Free Graph Contrastive Learning of Invariant-Discriminative Representations. TNNLS 2023.

[4] MA-GCL: Model Augmentation Tricks for Graph Contrastive Learning. AAAI 2023.

**Questions:**

Could you provide the experimental settings for reproducing the results (e.g., hyper-parameter configurations for the model and baselines)?

---

> ### Author Rebuttal · Authors · 2024-08-05
>
> We genuinely appreciate the time and effort you dedicated to thoroughly reading our paper. Our code is uploaded in the **Supplementary Material**, with optimal hyperparameters in the config file. Set "use_best_cfg = True" to reproduce our results. Specific training hyperparameters and dataset details are in **Tables 5 and 6 in Appendix A**. If you have any further questions about reproducing the results, please do not hesitate to contact us. We are more than happy to address any further concerns you may have.
>
> ### W1： More discussion on Novelty
> Thank you for your insightful questions. We first highlight a notable difference: DiGGR uses a different learning strategy compared to DGCL [1] and IDGCL [2]. They use a contrastive learning framework requiring negative samples and complex data augmentation methods ([1, 2] tested four different augmentation methods). In contrast, our model employs Graph Mask Autoencoder, with masking the input node as the only pretext task, eliminating the need for explicit negative samples.
>
> Due to character limitations, we thoroughly discussed the differences between DiGGR and [1-2] in the **global rebuttal** section. Please refer to that section for details.
> In short, the differences primarily arise from three aspects:  the general framework we use, the factor learning method we designed, and the probabilistic techniques specifically tailored for the generative model. We will also incorporate this comparison into future revisions of the paper.
>
> ### W2: recent or related contrastive baselines
> Grateful for you valuable suggestion. In the subsequent revision of our paper, we will include additional disentangled methods based on contrastive learning. For now, we have re-listed our tables below, showing only the results reported in the paper for [1-4].
>
> | Methods |  IMDB-B  |IMDB-M |MUTAG|NCI1|REDDIT-B |PROTEINS |COLLAB |
> |-----|------|---------|---------|-----|------|------|------|
> | DGCL        |  75.9 ± 0.7   | 51.9±0.4      | 92.1±0.8      | 81.9±0.2      |  91.8±0.2     |  76.4±0.5     | 81.2±0.3      |
> | IDGCL       | 76.1 ± 0.2    | 52.3  ±  0.4  | 92.5 ± 0.6    | 82.4  ± 0.3   | 91.9  ± 0.3   |  77.1 ± 0.2   | 81.3  ± 0.3   |
> | DiGGR       | **77.68 ± 0.48**  | **54.77 ± 2.63**  | 88.72 ± 1.03  | 81.23 ± 0.40  | 88.19 ± 0.28  | **77.40 ± 0.05**  |  **83.76 ± 3.70** |
>
> |Methods | Cora  | Citeseer  | Pubmed    | PPI      |
> |-|-|-|-|-|
> | MA-GCL      | 83.3 ± 0.4    | 73.6 ± 0.1    | 83.5 ± 0.4    | -            |
> | DiGGR       | **84.96 ± 0.32**  | **73.98 ± 0.27**  | 81.30 ± 0.26  | **78.30 ± 0.71** |
>
> 1. Due to large variations across different validation folds, we agree for each individual dataset the improvement might not appear that significant (not only to us, this observation applies to almost all previous state-of-the-art methods, e.g., for node classification, SeeGera[5] achieves better accuracy than GraphMAE[6] 0.1\% on the Cora dataset; for graph classification, IDGCL[2] outperforms DGCL[1] by 0.2\% on the IMDB-B dataset.).
> However, we would like to highlight that our model employs a generative self-supervised learning framework. Compared to contrastive learning-based methods [1-4], it achieves comparable results across 11 datasets, with **7 of them reaching optimal performance**. This is unlikely to occur by chance and thus can be used to verify its appealing performance.
>
> 2. We chose generative self-supervised learning method, specifically a Graph Masked Autoencoder (GMAE) as the foundational framework,  and incorporated disentangled learning into GMAE to help bridge the gap between generative and contrastive methods in node classification and graph classification task. As demonstrated in Tables 1 and 2 in our paper, **DiGGR  outperformed most methods based on generative frameworks**. We aim to further reduce the performance gap between generative SSL and contrastive methods in the graph domain, capitalizing on its advantage of not requiring complex, task-specific pretext tasks, and to extend its applications in graph-related fields.
>
> ### W3: Larger Datasets
> Following your advice, we tested our model on the ogbn-arxiv dataset. Since our model involves the reconstruction of the adjacency matrix, to further save computational costs on larger graphs, we adopted the scalable sampling method from [7], computing only a portion of the adjacency matrix at a time. Our experimental setup was as follows: Factor Number $K=4$, learning rate = 0.0005, training Epoch = 600. All other experimental settings and environments were kept consistent with those described in the main text. The results are shown below:
>
> |      Methods         |    ogbn-arxiv    |
> |:-------------:|:----------------:|
> | GraphMAE   |   71.75 ± 0.17   |
> | GraphMAE2  |   71.95 ± 0.08   |
> | DiGGR         | **72.12 ± 0.08** |
>
> It can be seen that DiGGR outperforms GraphMAE and GraphMAE2 on the ogbn-arxiv dataset, further validating the effectiveness of our approach. In the future, we will consider extending our model to other larger benchmarks. For extremely large graphs, we plan to use techniques such as PPR-Nibble[8] to generate relatively smaller local subgraph clusters. We believe that DiGGR has the potential to handle large-scale data and we are continuously working on this.
>
> [1] Disentangled contrastive learning on graphs. NeurIPS 2021
>
> [2] Disentangled graph contrastive learning with independence promotion. TKDE 2022
>
> [3] Augmentation-Free Graph Contrastive Learning of Invariant-Discriminative Representations. TNNLS 2023.
>
> [4] MA-GCL: Model Augmentation Tricks for Graph Contrastive Learning. AAAI 2023.
>
> [5] Seegera: Self-supervised semi-implicit graph variational auto-encoders with masking. WWW 2023
>
> [6] Graphmae: Self-supervised masked graph autoencoders. KDD 2022
>
> [7] Fastgae: Scalable graph autoencoders with stochastic subgraph decoding. Neural Networks 142 (2021): 1-19
>
> [8] Local graph partitioning using pagerank vectors. (FOCS'06). IEEE, 2006

---

### Author Rebuttal · Authors · 2024-08-06

## **Global Response**
We thank all the reviewers for their valuable suggestions on our paper. We will first address a common issue: **Differences with Previous Work**.
We have carefully reviewed the paper [1-5] you provided, and while it is true that both our method and them utilize KL divergence to optimize the latent factor, we believe our work has fundamental differences with them.

While these methods [1-5] have proven effective in graph contrastive learning, directly applying them to mask graph autoencoders poses challenges. Specifically, as Table 3 illustrates, using these methods (Non-probabilistic Factor Learning) in mask graph autoencoders hinders the learning of meaningful factors and leads to poor performance improvements. This difference may arise because contrastive learning, aided by data augmentation technology, is more likely to facilitate the convergence of latent factor models, whereas generative self-supervised learning, which relies solely on masking techniques, may struggle to achieve such convergence.

In addition to experimental findings, we further clarify the following three key aspects, starting with the model itself:

1. Compared with prior works [1-5], our approach treats the latent factor as a distribution rather than a point vector. In [1-5], a prototype-based method is employed where the node's hidden representation is directly mapped to a simplex point vector via a softmax function. In contrast, our method models the latent factor $Z$ using a Gamma distribution. Leveraging the sparsity properties of the Gamma distribution, our model extracts more distinctive latent factors. To validate this claim, we conducted quantitative experiments. The table below shows the normalized mutual information (NMI) between the top two latent factors extracted by different models."

|                NMI               |  COLLAB  |  IMDB-M  |
|:--------------------------------:|:--------:|:--------:|
| Non-Probabilistic Factor Learning |   1.00   |   0.94   |
|               DiGGR              | **0.35** | **0.24** |

A smaller NMI indicates lower similarity between the latent factors, meaning that the extracted factors are more distinguishable. It can be observed that the NMI between the latent factors in DiGGR is lower than that of the non-probabilistic methods, suggesting that DiGGR can extract more distinguishable latent factors and achieve better model convergence.

2. In DiGGR, learning latent factors considers both the structural information of the graph and task-related information. However, previous works [1-4] primarily rely solely on task-related information for learning latent factors. Specifically, as shown in Equation 7 of the paper:
$$
L_z = E_{q(Z|A,X)}[\ln{p(A|Z)}] - \sum_{u=1}^NE_{q(z_u|A,X)}[\ln \frac{q(z_u|A, X)}{p(z_u)}]
$$
Where $L_z$ includes a reconstruction term for the adjacency matrix $\ln p(A|Z)$, addressing the graph's structural information, which distinguishes our approach from previous works [1-4].

3. Both DiGGR and [5] employ graph factorization, yet they differ notably in their treatment of latent factors. DiGGR models these factors using a Gamma distribution, leveraging its non-negative and sparse characteristics. This approach aligns DiGGR with Bayesian non-negative matrix factorization principles, known for enhancing feature disentanglement, as highlighted in literature [6]. In contrast, [5] adopts a Gaussian distribution for latent factors, allowing for the inclusion of negative values during sampling. This departure from non-negativity, as discussed in [6], potentially undermines the benefits observed in DiGGR. To validate this assertion, we extracted latent factors from both methods and evaluated their performance on downstream tasks using datasets such as IMDB-BINARY and MUTAG. The outcomes are as follows:

|         |    IMDB-BINARY   |       MUTAG      |
|:-------:|:----------------:|:----------------:|
| PGD-VAE |   54.37 ± 0.21   |   84.06 ± 0.52   |
|  DiGGR  | **70.79 ± 0.34** | **85.12 ± 0.64** |

It can be observed that the latent factors from DiGGR outperform those from [5] on downstream tasks. This further underscores the distinctiveness and effectiveness of our latent factor learning method.

In summary, we believe that DiGGR differs from [1-5] in several key aspects. In the subsequent revisions of our paper, **we will revisit and discuss the work of [1-5] in the related work section**. Additionally, we will include a discussion section in the methodology part of the paper to help readers better understand the distinctions between DiGGR and these prior methods.

[1] Disentangled contrastive learning on graphs. NeurIPS 2021

[2] Disentangled graph contrastive learning with independence promotion. TKDE 2022

[3] Disentangled graph collaborative filtering. SIGIR 2020.

[4] Disentangled Graph Convolutional Networks. ICML 2019.

[5] Deep Generative Model for Periodic Graphs. NeurIPS 2022.

[6] Learning the parts of objects by non-negative matrix factorization. nature 401.6755 (1999): 788-791.

---

### Decision · Program_Chairs · 2024-09-25

**Decision:**

Reject

**Comment:**

The scores of this paper are mixed. After reading the paper and the authors' rebuttals, I decided to reject this paper due to its incremental contributions to methodology and performance.

Firstly, as the reviewers pointed out, there are many existing disentangled graph representation learning methods, while few of them are treated as baselines in this work, which makes the novelty of the proposed method questionable. Although the authors highlighted the differences between the proposed method and the existing ones, the claims were not verified by sufficient experiments. Especially, a comprehensive comparison with the method in "Deep Generative Model for Periodic Graphs. NeurIPS 2022" is necessary because the main difference between the proposed method and this one is the distribution assumption of latent factors.

Secondly, considering the change in the averaged result and the corresponding standard deviation, the performance improvements caused by the proposed method are incremental in most situations. The additional results shown in the rebuttal phase did not resolve this concern successfully. In addition, according to the figures shown in the paper and the pdf file provided in the rebuttal phase, the disentangled latent factors seem to imply clustered nodes, which is questionable for representing heterophilic graphs (and in my opinion, it might be the reason for the incremental performance improvements, and I strongly suggest the authors add analytic experiments to verify whether such a correlation exists and what a kind of impact it has on the model performance).

In addition, there are many typos in the paper, e.g., the "grpah" in the titles of Sections 3.3 and 3.3.1.

In summary, although no comprehensive discussions happened in the rebuttal phase (As AC, I am sorry to the authors, but I have tried my best to remind the reviewers), based on the analysis above, I suggest the authors polish this work and resubmit it in the future.